



# Multiproxy analyses of multiple firn cores from coastal Adélie Land covering the last 40 years

Titouan Tcheng[1], Elise Fourré[1], Christophe Leroy-Dos-Santos[1], Frédéric Parrenin[2], Emmanuel Le Meur[2], Frédéric Prié[1], Olivier Jossoud[1], Roxanne Jacob[1], Bénédicte Minster[1], Olivier Magand[3], Cécile Agosta[1], Niels Dutrievoz[1], Vincent Favier[2], Léa Baubant[1], Coralie Lassalle-Bernard[1], Mathieu Casado[1], Martin Werner[4], Alexandre Cauquoin[5], Laurent Arnaud[2], Bruno Jourdain[2], Ghislain Picard[2], Marie Bouchet[1], Amaëlle Landais[1]

[1]Laboratoire des Sciences du Climat et de l'Environnement, LSCE-IPSL, CEA-CNRS-UVSQ/IPSL, Saclay, France

[2]Institut des Géosciences de l'Environnement –CNRS-UGA-IRD-INP, Grenoble, France

[3]Observatoire des Sciences de l'Univers de la Réunion (OSU-Réunion), UAR 3365, Université de la Réunion, CNRS, Météo-France, IRD, Saint-Denis, France

[4]Alfred Wegener Institute for Polar and Marine Research, Bremerhaven, Germany

[5]Institute of Industrial Science, The University of Tokyo, Kashiwa, Japan

*Correspondence to*: Titouan Tcheng (titouan.tcheng@lsce.ipsl.fr)

**Abstract.** Water stable isotope signals recorded in shallow firn cores are essential to constrain the variations of climate and atmospheric water cycle over the past decades to centuries. However, deposition and post-deposition effects add additional signal, often referred to as stratigraphic noise, to the isotopic signal. One way to reduce the local stratigraphic noise is to combine several firn cores at the same location.

Here, we study the water isotopic composition and chemical records from 9 firn cores (20 to 40 m depth) drilled in 2016 at 3 sites (D47, Stop5 and Stop0) with high accumulation rates (~200 mm w.eq ·yr⁻¹) along a transect between the coast and the plateau in Adélie Land in Antarctica (100 to 385 km from the coastal station Dumont d'Urville). Each core covers at least the period from 1979 to 2016 and the high-resolution measurements permit to capture the seasonal variations in both chemical and isotopic records. At each site, similarities in the nssSO₄ and δ¹⁸O variations between the different cores were used to combine the three isotopic records into a single stacked isotopic curve, thereby enhancing the signal-to-noise ratio. At two sites, we find a good agreement when comparing the water isotopic profiles recovered from the stacked records to those obtained as modeling output from virtual firn cores calculated using the two isotope-enabled atmospheric general circulation models, ECHAM6-wiso and LMDZ6iso over the period 1979-2016. At the very windy site of D47, building a coherent signal from the 3 individual cores is not possible because the isotopic and impurities signals are much more affected by stratigraphic noise. This study confirms that, even if the benefit of stacking is limited at very windy sites, combining several cores is of primary importance to faithfully reconstruct water isotope variability at one site and further investigate how much climate signal is actually preserved in coastal cores.



## 1 Introduction

Antarctica is a key region for studying current climate change as future sea-level rise is closely linked to loss of the Antarctic ice sheet (Klose et al., 2024 and references therein). The link between future global temperature and Antarctic climate evolution relies on models that require validation based on observations (Bracegirdle et al., 2019). However, due to the challenging accessibility of the Antarctic continent, direct field observations, which are complementary to satellite observations, have been and remain limited. They are primarily obtained at research station locations and by automatic weather stations over recent decades. This makes the use of paleodata essential for reconstruction of climate variability and surface mass balance in Antarctica at the regional scale (Stenni et al., 2017).

Coastal Adélie Land is a region of Antarctica characterized by relatively high mean accumulation rates (200 to 400 mm w.eq·yr$^{-1}$ vs ~30 mm we.eq·yr$^{-1}$ for instance at the Dome C site on the plateau) and strong katabatic winds which deeply affect climate dynamics, the surface mass balance and the atmospheric hydrological cycle (Wendler et al., 1997; Magand et al., 2007; Goursaud et al., 2017; Le Meur et al., 2018; Davrinche et al., 2024). As an example, katabatic winds may be responsible for the sublimation of up to 35 % of the total precipitation (Pettré et al., 1993; Grazioli et al., 2017). Strong winds also impact snow erosion and deposition (Amory, 2020), leading to significant local (Poizat et al., 2024) to regional scale redistribution (Agosta et al., 2019). Finally, the area is marked by large variations of the sea-ice extent (Crosta et al., 2021) and by a strong cyclogenesis due to the presence of both katabatic winds and cyclones dissipation to the west (Bromwich et al., 2011).

Water isotopes are a useful tool to reconstruct temperature variations from measurements on shallow and deep ice cores to extend climatic reconstructions beyond meteorological records in the past (Jouzel et al., 2007; Cuffey et al., 2016; Stenni et al., 2017; Casado et al., 2023; Jones et al., 2023). First reconstructions were based on the spatial relationship observed between the evolution of water isotope ratios expressed in terms of $\delta^{18}$O or $\delta$D and temperature (Lorius and Merlivat, 1975; Masson-Delmotte et al., 2008) while the most recent ones have taken into account some aspects of the atmospheric water cycle such as the precipitation intermittency and the origin of the precipitation by using information provided by atmospheric general circulation models (AGCM) equipped with water isotopes (Goursaud et al., 2017; Kino et al., 2021). The recent increase in number of water vapor isotopic observation series in Antarctica and comparison to model outputs at several stations (Casado et al., 2016; Ritter et al., 2016; Bréant et al., 2019; Bagheri-Dastgerdi et al., 2021; Leroy-Dos Santos et al., 2021) strengthen confidence in the use of isotope-enabled AGCMs to  interpret water isotopic series in ice cores. Coastal Adélie Land is a particularly relevant area for such study, with the availability of a long-term water isotopic series for water vapor and precipitation (continuously since January 2019) at the Dumont d'Urville station comparing well with isotope-enabled AGCM outputs (Leroy-Dos Santos et al., 2023) .

The water isotopic composition archived in firn and ice cores is not only influenced by variations in temperature, precipitation intermittency and atmospheric water cycle, but also by several post-deposition effects such as sublimation or surface hoar formation, wind redistribution of surface snow according to surface relief and diffusion within snow of high open porosity (Langway, 1967; Johnsen et al., 2000; Town et al., 2008; Touzeau et al., 2018; Zuhr et al., 2023). In particular, coastal Adélie Land is strongly affected by wind-induced redistribution of snow because of the strength of the katabatic winds (Amory, 2020). This process induces variability in the water isotopic records in firn and ice cores which cannot directly be linked to large scale climatic variability. This



variability makes the so-called stratigraphic noise, which complicates the retrieval of climate signal from water isotopes (Fisher et al., 1985; Hirsch et al., 2023).

Our study aims at (1) reconstructing the isotopic composition of precipitation from firn cores in coastal Adélie land with a reduced impact of stratigraphic noise and (2) assessing how much climatic information can be retrieved from water isotopic records in these firn cores. We analyse a unique set of 9 shallow firn cores drilled at 3 sites in the coast-to-plateau transition zone in Adélie Land (Figure 1). The drilling sites were chosen so that accumulation rate was high enough to be able to capture annual layers (minimum of 200 mm we.eq·yr$^{-1}$ according to Frezzotti et al., 2007). Section 2 details the field campaign, measurements and methodologies to establish the chronology using the Paleochrono software and build virtual firn cores from AGCM models. In section 3, we present the measured records, the common chronology built using constraints from impurities and $\delta^{18}O$ data, Ground Penetrating Radar (GPR) measurements as well as beta counting and gamma spectroscopy. We then compare the stacked $\delta^{18}O$ records at each site with virtual firn cores (VFCs) derived from the isotope-enabled AGCMs ECHAM6-wiso and LMDZ6iso. In section 4, we investigate the possibility to retrieve climatic information from our stacked cores. We summarize the main findings and propose directions for future research in section 5.

## 2 Material and method

### 2.1 Field work and sampling sites

The firn cores analysed in this study were obtained during the field campaign of the ASUMA project (Assessing SUrface Mass balance of Antarctica) which aimed at quantifying the recent variations of surface mass balance along a transect in Adelie Land. The field campaign was carried out from December 2016 to January 2017 and was led by an expedition involving the French institutes IGE (Institut des Géosciences de l'Environnement) and IPEV (Institut Polaire Français Paul-Emile Victor). 25 shallow firn cores, ranging from 20 m to 40 m long, were drilled along a 1371 km-long loop (Figure 1) starting from the Italian–French Cap Prud'Homme station, located at sea level 5 km away from Dumont D'Urville on the continent and ascending to 2416 m above sea level at Stop0 point (385 km from the Dumont D'Urville station).



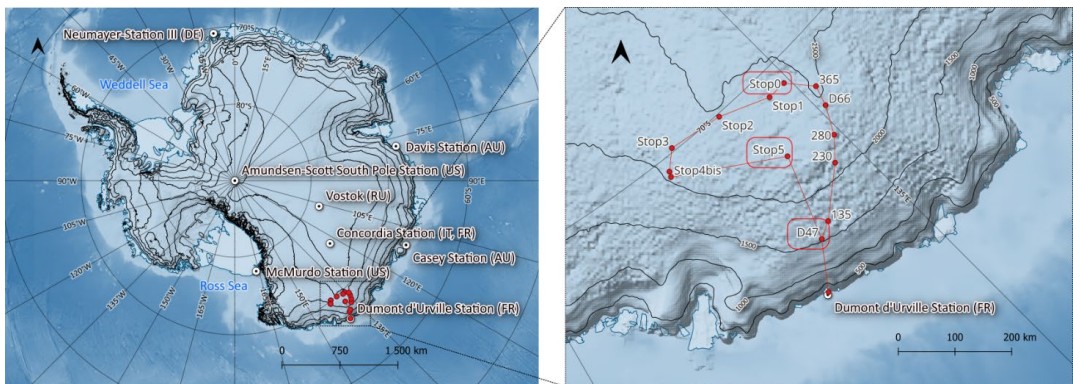


**Figure 1: Map of Antarctica and the ASUMA raid. Red dots correspond to the drilling points. The sites considered in this study are D47, Stop5 and Stop0. Isohypses are derived from CryoSat-2-derived elevation model of Antarctica (Helm et al., 2014) with Quantarctica package (Matsuoka et al., 2021). Cap Prud'homme mentioned in the main text can not be distinguished from the Dumont d'Urville station at the scale of the maps.**


The present study focuses on 9 firn cores that were drilled at 3 sites: D47, Stop5 and Stop0 (Figure 1, Table 1). The cores were extracted using a portable electromechanical drilling system (Ginot et al., 2002). After bulk density measurements, the cores were sealed in polyethylene bags, stored in clean isothermal boxes and shipped to IGE cold room facilities in Grenoble, France.

Ground penetrating radar (GPR) data between the sites were collected using a MALÅ® ProEx system fitted with a 100 MHz rough-terrain antenna, towed by one of the tractors from the traverse. The distance between the transmitter and the receiver was consistently maintained at 2.2 m. This setup was similar to the one employed on a traverse between Dome C and Vostok (Le Meur et al., 2018). Triggering was set at 2 second intervals, resulting in a radar trace every 8 m, based on the convoy's average speed of about 14 km·h$^{-1}$. A time window of roughly

1.5 µs was applied, which allowed for a depth of investigation of about 125 m, given a 128-fold stacking and an average wave velocity of 0.2 m·ns$^{-1}$. The resulting waveform was then sampled at a rate of about 10 times the nominal center frequency of the antenna (i.e $10 \times 10^8$ Hz = 1MHz) thereby providing some 1500 samples per trace. A GNSS receiver mounted on the vehicle recorded the geographic position of each trace along the survey profiles. Pre-processing during acquisition included 128-fold stacking to enhance the signal-to-noise ratio. Post-processing

steps involved a time-zero corrections, a zero-phase low-cut filter (devow) to eliminate continuous direct currents, and an 'energy decay' gain to counteract signal attenuation from volumetric spreading. Band-pass filtering was replaced by spatial averaging over 25 to 50 traces, which was deemed more efficient than traditional finite impulse response (FIR) filters due to the large dataset (~150,000 traces). Time-to-depth conversion was performed by migrating radargrams using a vertical velocity profile for the radar wave.

**Table 1: Details of the drilling sites of the 9 firn cores retrieved in Adélie Land in December 2016 and January 2017 and presented in this study.**

| Site | Coordinates | Altitude | Distance from Dumont d'Urville (km) | Drilling date | Core name | Depth (m) |
|------|-------------|----------|-------------------------------------|---------------|-----------|-----------|
|      |             |          |                                     |               |           |           |



| | | (m.a.s.l) | | | | |
|---|---|---|---|---|---|---|
| **Stop0** | 69.6359° S, 135.2796 °E | 2416 | 385 | 9-12/12/2016 | ASUMA2016_6 | 20.37 |
| | | | | | ASUMA2016_7 | 44.60 |
| | | | | | ASUMA2016_8 | 20.43 |
| **Stop5** | 68.7473° S, 137.4428° E | 2317 | 255 | 26-27/12/2016 | ASUMA2016_19 | 19.98 |
| | | | | | ASUMA2016_20 | 20.00 |
| | | | | | ASUMA2016_21 | 32.73 |
| **D47** | 67.3989° S, 138.7094° E | 1516 | 100 | 2-3/01/2017 | ASUMA2016_23 | 20.98 |
| | | | | | ASUMA2016_24 | 40.47 |
| | | | | | ASUMA2016_25 | 17.59 |

### 2.2 Measurements

Water stable isotope composition of the core ASUMA2016-7 was measured by laser spectrometry
(Picarro L2130-i) on discrete samples at a 4 cm resolution at the Laboratoire des Sciences du Climat et de
l'Environnement (LSCE) corresponding to a temporal resolution of 0.9 to 1.4 month. The estimated 1-sigma
uncertainties are 0.2 ‰ for $\delta^{18}O$ and 0.7 ‰ for $\delta D$ (Grisart et al., 2022). All the other cores were measured using
a Continuous Flow Analysis (CFA) facility paired with a Picarro L2130-i water vapor analyzer and a home-made
vaporiser. We used a setup based on a similar principle as the one described in Dallmayr et al. (2016, 2025a) and
presented in Petteni et al. (2025, in discussion) to melt 30 mm x 30 mm sticks at an average rate of 2.5 cm·min$^{-1}$.
Only the water from the inner clean 18 mm x 18 mm part of the stick was kept for analysis, and split to feed the
Picarro spectrometer on one side and a fraction collector on the other side. Isotopic calibrations were conducted at
the beginning and end of each CFA day using 3 in-house standards themselves calibrated against Vienna Standard
Mean Ocean Water (VSMOW) and Standard Light Antarctic Precipitation (SLAP): Ross ($\delta^{18}O$ = -18.63 ± 0.05
‰, $\delta D$ = -144.8 ± 0.7 ‰), NEEM ($\delta^{18}O$ = -32.89 ± 0.05 ‰, $\delta D$ = -254.1 ± 0.7 ‰) and Adélie ($\delta^{18}O$ = -40.55 ±
0.05 ‰, $\delta D$ = -321.0 ± 0.7 ‰). By running check standards in some CFA runs, we concluded that the overall
uncertainties given for discrete measurements also hold for CFA isotopic measurements.

The fraction collector was set to collect 3 mL samples, which corresponds to a depth resolution between
20 and 30 mm. Major ions concentrations ([Na$^+$], [Cl$^-$] and [SO$_4^{2-}$]) were measured at LSCE using two Dionex ion



chromatography systems working in parallel (ICS-5000+). For cations, separation was obtained using a CG16-4μm (2x50 mm) guard column and a CS16-4μm (2x250 mm) separation column, with isocratic runs of 30 mMol MSA as eluent at 0.17 ml·min$^{-1}$. For anions, the eluent was 55 mMol KOH at 0.3 ml·min$^{-1}$ and the system was equipped with an AG28-4μm (2x30 mm) guard column and AS28-4μm (2x150 mm) separation column. Both systems were running with Dionex DRS600 (2 mm) dynamically regenerated suppressors. By calculating the

pooled standard deviation over a panel of 93 samples remeasured with a different set of standards, we estimated a 1-sigma uncertainty of 1.0 ppb, 1.2 ppb and 0.8 ppb for [Na$^+$], [Cl$^-$] and [SO$_4^{2-}$] respectively. The non-sea-salt SO$_4^{2-}$ (nssSO$_4$) was calculated following the classical approach: nssSO$_4$ = [SO$_4^{2-}$] – 0.25[Na$^+$] (Abram et al., 2013; Nardin et al., 2021). This relationship assumes a [SO$_4^{2-}$]/[Na$^+$] ratio value of 0.25 for marine aerosol and a negligible fractionation.

Additionally, artificial radionuclides resulting from atmospheric thermonuclear tests carried out from the 1950s to 1980s were deposited in Antarctica after being transported in the upper troposphere and stratosphere, creating unambiguous chronostratigraphic markers in the firn cores, peaking at years 1955±1 and 1965±1 AD (Jouzel et al., 1979; Pourchet et al., 1983, 1997, 2003; Le Meur et al., 2018). These layers were identified via the detection of $^{90}$Sr, $^{241}$Pu (deduced from $^{241}$Am analysis) and $^{137}$Cs. To perform this analysis, firn cores from Stop0

and D47 were processed and cut for artificial radioactivity measurements applying a method developed by Delmas and Pourchet (1977): core sections (100-250 g) were melted, weighted, acidified and filtered through ion exchange resin papers (MN 616 LSA-50 and LSB-50 strongly acidic cation and basic anion exchange resins, respectively) where all the radionuclides are trapped. Filters were first analyzed at IGE semi-underground laboratory by an alpha/beta Berthold LB-790 low-noise counter (Magand, 2009). A more detailed and targeted detection of the

$^{137}$Cs and $^{241}$Am radionuclides was then performed using very low background gamma spectrometry in the Modane deep underground laboratory (LSM) (Hodák et al., 2019).

### 2.3 Firn core synchronization and dating

    The Paleochrono software serves as a probabilistic dating tool, akin to Datice and Icechrono1, with

enhanced mathematical, numerical, and programming capabilities (Lemieux-Dudon et al., 2010; Parrenin et al., 2024). It has been originally designed to establish a unified and optimized chronology for archives from various paleoclimatic sites. It has then largely been used for ice core dating, both for deep ice cores (Bouchet et al., 2023; Oyabu et al., 2022) or for shallow firn cores (Oyabu et al., 2023) as also done in the present study. When applied to ice cores, Paleochrono relies on Bayesian inference for shaping the underlying background scenario of

accumulation rate ($A$) and thinning of annual ice layers ($\tau$).

    To quantify the credibility of the background scenario, the accumulation function and chronological information (e.g., dated horizons, stratigraphic tie-points between cores) are expressed as a probability density, assumed to be Gaussian. The coherent age scale construction then relies on the Least Square optimization method and results in the best compromise between changes of the background scenarios within their associated

uncertainties (imposed by the user) and respect of the chronological observations within their associated uncertainties.



Here we assigned a constant prior accumulation for the period covered by our records using available information from radionuclides-based dating, Ground Penetrating Radar isochrones between Stop5 and D47, surface accumulation stakes or Modèle Atmosphérique Régional (MAR) estimations (Agosta et al., 2019).
Although some of these methods (beta counting and gamma spectrometry, stake measurements) provide accurate long-term average estimates of accumulation rate, we deliberately associate them with a relatively high uncertainty (50 to 200%) to allow Paleochrono to produce a variable accumulation rate in time based on the alternative chronological and stratigraphic constraints. Given that the present study focuses on firn cores, ice thinning is not a relevant parameter. Instead, the density evolution vs. depth was measured on each firn core and is implemented
in the background scenario with negligible associated uncertainty.

The chronological constraints for our study are the followings. When the data are non-ambiguous, the beta counting and gamma spectroscopy provides dated horizons. We also incorporated dated intervals for each core, assessed by counting the number of seasonal peaks over regular 2-meter intervals. Peaks and cycles were identified synchronously in both water isotopes and impurity records which supports recording of annual layers in
the firn cores (see section 3). Note that for some sections only water isotopes were used for layer counting since there are some gaps in the impurity records especially for Stop0 (even no data on one of the 3 cores). When counting annual layers, we computed the uncertainty by adding 0.5 yr each time the identification of a seasonal layer was doubtful. This approach follows the one used in Rasmussen et al. (2006) to build the GICC05 chronology and applied in the common Antarctic chronology exercise by Lemieux-Dudon et al. (2015).

Finally, we identified multiple stratigraphic tie-points in pairs of ice cores at each site which permit aligning the three records using both isotopic and chemical data. The selection of the stratigraphic tie-points between cores is based on the visual identification of common patterns or peaks by comparison of the chemistry and water isotope records. Depending on the confidence we have in the pattern correspondence between different cores, we assigned an uncertainty from 0.2 yr (unambiguous matching) up to 2 yr (when matching is disputable)
for each tie-point. These choices are reasonable but subjective and constitute a documented limitation of the Paleochrono approach (Parrenin et al., 2024).

### 2.4. Virtual Firn Core built from ECHAM6-wiso and LMDZ6iso outputs

To help with the interpretation of our water isotopes records, we compared them to a simulated $\delta^{18}$O
signal in firn cores inferred from isotope-enabled AGCM outputs and hereinafter called Virtual Firn Cores (VFC). It follows the general approach of Sime et al. (2011) revisited by Casado et al. (2020). We use $\delta^{18}$O in precipitation and precipitation amount timeseries outputs of two isotope-enabled AGCMs, ECHAM6-wiso and LMDZ6iso extracted from the grid cells of our sites of interest. The precipitation amounts of ECHAM6-wiso and LMDZ6iso were scaled so that the average of the modeled accumulation rate is equal to the mean accumulation inferred at
each drilling site. The effect of diffusion on the firn isotopic profile is calculated using the classical diffusion model after Johnsen et al. (2000) with a diffusion length calculated from the density profile, the accumulation rate and the mean annual surface temperature.

ECHAM6-wiso is the isotopic version of the ECHAM6 AGCM (Stevens et al., 2013), developed to explicitly simulate isotopic variations in the atmospheric hydrological cycle. The implementation of water isotopes



in ECHAM6-wiso has been previously described by Cauquoin et al. (2019), and has since undergone several updates to improve its consistency with recent observations of water isotope behavior (Cauquoin and Werner, 2021). These updates include the incorporation of the isotopic composition of snow over sea ice, a revised supersaturation equation, and the assumption of wind-speed independence for kinetic fractionation factors during oceanic evaporation. The orbital parameters and greenhouse gases concentrations used in the ECHAM6-wiso

simulation were set to the values corresponding to the modeled year. We have used daily values of ECHAM6-wiso model outputs from a simulation at high spatial resolution (0.9° horizontal resolution and 95 vertical levels) nudged to European Centre for Medium-Range Weather Forecasts (ECMWF) ERA5 reanalysis (Hersbach et al., 2020). The ECHAM6-wiso 3D fields of temperature, vorticity and divergence as well as the surface pressure field were nudged toward the ERA5 reanalysis data every 6 h. A detailed description and evaluation of the ECHAM6-

wiso simulation can be found in Cauquoin and Werner (2021).

LMDZ6iso (Risi et al., 2010) is the isotopic version of the AGCM LMDZ6 (Hourdin et al., 2020). We used LMDZ6iso version 20231022.trunk with the physical package NPv6.3, nearly identical to the atmospheric setup of IPSL-CM6A (Boucher et al., 2020) used for phase 6 of the Coupled Model Intercomparison Project (CMIP6, Eyring et al., 2016). The simulation was performed with the standard low resolution (LR) horizontal grid

(2.0° in longitude and 1.67° in latitude, 144x142 grid cells) and 79 vertical levels, with the first atmospheric level located around 10 m above ground level. The LMDZ6iso 3D fields of temperature and wind are nudged toward the 6-hourly ERA5 meteorological reanalysis data (Hersbach et al., 2020) with a relaxation time of 3 h, except below the sigma-pressure level equivalent to 850 hPa above sea level, where no nudging is applied in order to let the model's physics and dynamics to be expressed in the boundary layer. Surface ocean boundary conditions are

derived from monthly mean sea surface temperature and sea-ice concentration fields from the ERA5 reanalysis. The simulation, identical as in Dutrievoz et al. (2025), covers the period 1980-2017, from which we use daily outputs.

Leroy Dos-Santos et al. (2023) demonstrated that ECHAM6-wiso $\delta^{18}O$ outputs exhibit a clear correlation with the isotopic composition of precipitation and water vapor at Dumont d'Urville station. The same

performances have been made for LMDZiso by Dutrievoz et al. (2025). These findings highlight the model capacity to help in the interpretation of shallow firn cores.

### 3 Results

### 3.1 Isotopic variability ($\delta^{18}O$ and $\delta D$ records)

Figures 2 and S1 present the evolution of $\delta^{18}O$ over the 9 firn cores and Table 2 presents the main

characteristics of the water isotopic records. As expected, the average $\delta^{18}O$ and $\delta D$ values are similar for the 3 cores drilled at a given site. Yet, average values of $\delta^{18}O$ and $\delta D$ display different orders of magnitude between sites ranging from approximately -27 ‰ (D47) to -37 ‰ (Stop0) for mean $\delta^{18}O$ and -209 ‰ (D47) to -290 ‰ (Stop0) for mean $\delta D$. The decreasing average $\delta^{18}O$ and $\delta D$ values from D47 to Stop0 through Stop5 were expected due to the gradual rise in elevation, decrease in temperature and the increasing distance from the coast (resulting

in greater atmospheric distillation).

The observed cyclicity suggests that water isotopes record seasonal cycles with interannual variability in the amplitude. When comparing shallow cores from Stop5 or from Stop0, the isotopic records display similar





variability in the seasonal variations at neighboring depths. This is not the case at D47, where the three isotopic

records display only few common patterns (Figure 2).

**Table 2: Statistics of the stable water isotopes composition of the 9 firn cores and the stack at each site (section 3.4) calculated along the common length of each core, i.e the length of the shortest core (20.37 m, 19.98 m and 17.59 m for Stop0, Stop5 and D47 respectively). For the two longer cores of each site, values in brackets are computed over the whole core length.**

| δ¹⁸O (‰) | | | | | |
|---|---|---|---|---|---|
| **Site** | **Firn core** | **Min** | **Mean** | **Max** | **Max-min** |
| **Stop0** | **6** | -41.3 | -36.1 | -27.3 | 14.0 |
| | **7** | -41.0 (-41.0) | -36.1 (-36.6) | -25.6 (-25.6) | 15.4 (15.4) |
| | **8** | -40.7 (-41.3) | -36.5 (-36.6) | -28.7 (-28.2) | 12.0 (12.0) |
| | **Stack** | -39.9 | -36.7 | -27.7 | 12.2 |
| **Stop5** | **19** | -38.3 | -34.4 | -30.0 | 8.3 |
| | **20** | -38.0 (-38.6) | -34.0 (-34.0) | -27.8 (-27.1) | 10.2 (11.4) |
| | **21** | -38.8 (-39.2) | -33.9 (-34.0) | -28.6 (-34.0) | 10.24 (12.1) |
| | **Stack** | -37.4 | -34.1 | -30.1 | 7.3 |
| **D47** | **23** | -31.1 (-31.5) | -26.7 (-26.8) | -21.1 (-20.6) | 10.0 (10.9) |
| | **24** | -31.5 (-32.1) | -27.0 (-26.9) | -22.0 (-19.5) | 9.5 (12.6) |
| | **25** | -31.2 | -26.9 | -21.7 | 9.5 |
| | **Stack** | -30.8 | -26.9 | -22.2 | 8.6 |
| δD (‰) | | | | | |
| **Stop0** | **6** | -328.4 | -291.6 | -208.4 | 119.9 |
| | **7** | -329.7 (-329.7) | -291.7 (-289.5) | -195.7 (-195.7) | 134.0 (133.96) |
| | **8** | -324.4 (-325.5) | -288.1 (-288.8) | -219.0 (-217.8) | 105.3 (107.7) |
| | **Stack** | -318.5 | -289.9 | -213.5 | 105.0 |
| **Stop5** | **19** | -304.1 | -271.8 | -233.9 | 70.1 |
| | **20** | -303.0 (-303.6) | -268.8 (-268.8) | -215.6 (-214.8) | 87.4 (88.8) |
| | **21** | -312.7 (-313.2) | -267.8 (-267.9) | -226.3 (-217.0) | 86.5 (96.2) |
| | **Stack** | -299.8 | -269.5 | -234.2 | 65.6 |
| **D47** | **23** | -242.1 (-243.0) | -208.4 (-209.3) | -162.6 (-161.6) | 79.5 (81.4) |
| | **24** | -246.3 (-247.9) | -210.9 (-210.6) | -167. (-151.4) | 78.5 (96.5) |



| | 25 | -246.1 | -208.0 | -162.9 | 83.2 |
| | Stack | -240.2 | -209.1 | -168.8 | 71.4 |
| nssSO4 (ppb) | | | | | |
| Site | Firn core | Min | Mean | Max | Max-min |
| Stop0 | 6 | 0 | 16.9 | 76.3 | 76.3 |
| | 8 | 0 | 17.6 | 72.3 | 72.3 |
| Stop5 | 19 | 1.2 | 18.0 | 77.2 | 76.0 |
| | 20 | 3.6 | 22.0 | 84.0 | 81.4 |
| | 21 | 2.3 | 20.0 | 84.9 | 82.6 |
| | Stack | 4.3 | 19.7 | 58.5 | 54.2 |
| D47 | 23 | 0 | 20.3 | 124.7 | 124.7 |
| | 24 | 0 | 21.8 | 102.4 | 102.4 |
| | 25 | 0 | 23.2 | 106.1 | 106.1 |
| | Stack | 3.2 | 21.9 | 88.6 | 85.3 |


**3.2 Chemistry records**

For each core, nssSO$_4$ profiles (Figure 2 and S1) feature common patterns that are very similar to those in Na$^+$ or Cl$^-$ profiles (not shown, data available in open repository). As we aimed at using these records to provide stratigraphic tie-points, we only concentrated on nssSO$_4$ records in the following. The mean nssSO$_4$ values are
17.3, 20.0 and 21.8 ppb at Stop0, Stop5 and D47 respectively (Table 2). The nssSO$_4$ records show variations at the same cyclicity as that observed in the water isotopes records. In particular, we observe a clear correspondence in the nssSO$_4$ and water isotopes cycles at Stop5 and Stop0 (Figure 3). These cycles have a periodicity compatible with the annual amount of accumulated snow (Table 3) and both water isotopes and nssSO$_4$ show concomitant maxima supporting the fact that seasonal cycles are well recorded at Stop0 and Stop5 in both nssSO$_4$ and water
isotopes. Such annual cycles in both water isotope and impurity records have been identified in other sites with accumulation rates in the same order of magnitude like WAIS Divide ice core in West Antarctica (Jones et al. 2017, 2023). At D47, cycles are also observed in nssSO$_4$ but the correspondence between nssSO$_4$ and water isotopes is less clear which do not support a good record of the seasonal cycles at this site. Finally, neither the Pinatubo nor the Agung eruptions can be unambiguously identified in the nssSO$_4$ records. A slight increase in the
average value of nssSO$_4$ can still be observed between 10 and 12 m at Stop0 and Stop5 and between 7 to 12 m at D47 and may include the signature of the Pinatubo eruption which is visible over years 1992 and 1993 in some Antarctic ice cores (Cole-Dai and Mosley-Thompson, 1999; Emanuelsson et al., 2022; Sigl et al., 2013). Because, we do not find an unambiguous nssSO$_4$ signal to attribute to the Pinatubo eruption, no volcanic horizon was included in the Paleochrono runs.





**Figure 2: $\delta^{18}O$ and nssSO₄ records at ASUMA sites Stop0, Stop5 and D47. The labels on the right vertical axis corresponds to the core number. Tie-points used for Paleochrono are represented with solid black lines. Yellow and blue $\delta^{18}O$ curves were shifted by -10 ‰ and +10 ‰ respectively to improve readability. At Stop5 and D47, red and blue nssSO₄ curves were shifted by +80 and +160 ppb respectively. At Stop0, blue nssSO₄ curves was shifted by +80 ppb. Horizontal lines correspond to the section where year 1992 and 1993 were identified in the output chronology along with their dating uncertainties (dashed lines).**







**Figure 3: δ¹⁸O and nssSO₄ records between the depth levels 7000 mm and 9500 mm for 3 three cores, ASUMA2016_6 (Stop0), ASUMA2016_20 (Stop5) and ASUMA2016_25 (D47) showing synchronous maximums in nssSO₄ and δ¹⁸O for Stop0 and Stop5.**






### 3.3 Prior deposition scenario

Paleochrono requires a prior estimate of the accumulation rate to derive a chronology at each site. Table 3 compiles all the accumulation data available derived from beta counting and gamma spectrometry, stakes
measurements, and simulation outputs from the Regional Atmospheric Model (MAR, Gallée and Schayes, 1994) forced by the ERA5 reanalysis covering the period 1979-2020 (same simulation as in Davrinche et al., 2024). At Stop5 and Stop0, the different estimates are close to each other, within 10%. Our results neither depend on the exact value of the Paleochrono prior accumulation rate among the estimates mentioned above nor on the value of the uncertainty attached to this prior accumulation rate (a final value of 50% has been used for the final chronology
presented below for Stop0 and Stop5 while at D47 the uncertainty was set to 200% due to the high variability of the different estimates).

**Table 3: Prior estimates of accumulation rates (mm w.eq·yr⁻¹) for each of the three sites from different methods. From left to right (see text for more details): 1/ Identification of the 1955 and 1965 bomb peaks in the cores allows to derive**
**an average accumulation from 1955 (resp. 1965) to 2017. At Stop5, no radionuclides measurements are available but we report values from a nearby core in a former study. 2/ Stakes annually visited in the field provide an estimate of accumulation rate: at Stop0, a stake has been installed as part of ASUMA project in 2016. Around D47, stakes from the SAMBA network are recorded since 2004; we report the values from the two closest stakes from our drilling site, highlighting the highly regional and year-to-year heterogeneity. The mean and standard deviation of the annually**
**measured accumulations are given in the table. 3/ MAR outputs from a simulation covering the period 1979-2020 with uncertainty calculated following the method described in Agosta et al. (2019, Supplement S3). 4/ We used the two GPR reflectors that we were able to identify in D47 core (at 10.9 m and 13.4 m depth) and follow up to Stop5, in order to derive mean accumulation rate estimates from Paleochrono output scenario at Stop5 (approach and uncertainty calculation in Le Meur et al. 2018). 5/ The last column reports the average of the annual accumulations given in**
**Paleochrono output scenarios for the longest core at each site. At D47, the first (resp. second) number in this column is the accumulation rate obtained with a 30% (resp. 200%) uncertainty on the background scenario of accumulation rate.**

| Site | Beta-gamma derived mean accumulation rate (mm w.eq.yr⁻¹) | Nearby accumulation estimations from stakes | | | MAR accumulation rates | | | GPR reflectors derived mean accumulation rate | Mean accumulation rate from Paleochrono output (mm w.eq.yr⁻¹) |
|---|---|---|---|---|---|---|---|---|---|
| | | Mean value and standard deviation (mm w.eq.yr⁻¹) (time coverage) | Method | Stake-to-site distance | Mean value (mm w.eq.yr⁻¹) | Gridpoint coordinates and altitude | Grid cell center-to-site distance | | |
| **Stop0** | 220 ± 7 *(1965-2017; this study)*<br>221 ± 7 *(1955-2017; this study)* | 224 ± 17 *(2016-2020)* | Stake installed for ASUMA project | ~100 to 200 m | 213± 29 *(1979-2020)* | 69.7062° S, 135.0000 °E (2470 m.a.s.l) | ~13.8 km | | 231 *(1906-2017)* |
| **Stop5** | 260± 70 *(1955-1976) (from Pettré et al., 1986* | | | | 240 ± 30 *(1979-2020)* | 68.8047° S, 137.4346° E (2357 m.a.s.l) | ~6.3 km | | 243 *(1943-2017)* |



| | | | | | | | | | |
|---|---|---|---|---|---|---|---|---|---|
| | *at a ~19 km site)* | | | | | | | | |
| **D47** | *168 ± 4 (1965-2017; this study)* | *132 ± 104 (2004-2020)* | SAMBA network Stake 988 *(Updated from Agosta et al., 2012)* | ~1 km | *372 ± 32 (1979-2020)* | 67.4645° S, 138.4336° E (1632 m.a.s.l) | ~8.0 km | *157 ± 21 (10.9 m)* | 177-263 *(1858-2017)* |
| | *174 ± 10 (1955-2017; this study)* | *294 ± 155 (2004-2020)* | SAMBA network Stake 960 *(Updated from Agosta et al., 2012)* | ~2 km | | | | *159 ± 21 (13.4 m)* | |

At D47, we observe strong differences among the estimates of accumulation rate. The beta counting and
gamma spectrometry data suggest depth of $19.10 \pm 0.5$ m and $16.10 \pm 0.5$ m for years 1955 and 1965, but the ages
obtained from these techniques are less reliable than at Stop0 due to uncomplete gamma spectrometry data caused
by technical issues during data treatment. The two closest stake measurements differ by a factor of more than 2,
despite being only 2 km away from one another. This reflects the strong short scale variability of accumulation
rates in this region. We thus believe that the MAR output at the D47 grid point is not a faithful estimate of the
accumulation at the exact location of the drilling site.

We thus explore an alternative way to estimate the accumulation rate at D47. Two GPR reflectors
identified between Stop5 and D47 were identified at 17.90 m and 20.44 m at Stop5 and at 10.89 m and 13.38 m at
D47. Using the ages obtained at Stop5 from Paleochrono with much more faithful chronological constraints than
at D47 (40.2 and 46.1 yr), we deduce accumulation rates of $157 \pm 21$ and $159 \pm 21$ mm w.eq.yr$^{-1}$, with uncertainties
derived from isochrones calculated as in Le Meur et al. (2018) (Table 3). We used the average of $158 \pm 21$ mm
w.eq.yr$^{-1}$ as input for Paleochrono. These estimates are close to the estimates inferred from the beta counting and
gamma spectrometry. For this site, there is a huge difference between the values of the accumulation rate obtained
as output of paleochrono for sensitivity tests performed with uncertainties of 30 or 200% in the background
accumulation rate (Table 3).

**3.4 Construction of stack series on an age scale**





For each drilling site, we construct stack series on a coherent age scale using the Paleochrono tool together with dating and stratigraphic constraints following the method described in section 2.3. In addition to the absolute dating constraints from beta counting and gamma spectrometry on Stop0, dating constraints relies on annual layer counting in 2m-intervals. As discussed above, the annual layers are clearly visible in both the $nssSO_4$ and water isotopes records at Stop0 and Stop5, whereas it is more ambiguous at D47 hence leading to larger associated uncertainties in the Paleochrono input files (Table S1). For combining the different records, we use stratigraphic tie-points based on the resemblance of the patterns in water isotopes (Table S2) and $nssSO_4$ (Table S3). Because this approach is subjective, we performed several sensitivity tests using either only the $nssSO_4$ tie-points, only the water isotopes tie-points, or both the $nssSO_4$ and water isotopes tie-points. We then compared at each site the stacks obtained for different configurations (Figure 4): (1) on the original depth scale (i.e. without any dating constraint), (2) on an age scale without any stratigraphic tie-points between the cores, (3) on an age scale with only $nssSO_4$ tie-points, (4) on an age scale with only water isotope tie-points and (5) on an age scale with both $nssSO_4$ and water isotope tie-points.

For the different sites, we do not observe large differences in the stacked water isotope records after matching records of $nssSO_4$, water isotopes or both $nssSO_4$ and water isotopes. As expected, the uncertainty calculated by Paleochrono is the largest when there are no stratigraphic tie-points and is the largest for D47 where the confidence on layer counting is low (Figure S2). At Stop5, the 1 σ dating uncertainty decreases from ~2.5 years (no tie-point) to ~1.5 year (only $nssSO_4$ tie-points) or slightly less (water stable isotopes and $nssSO_4$ tie-points).

We have a few ways to check the validity of the chronology calculated by Paleochrono. First, we find a good agreement between the a priori and output accumulation rate at Stop5 and Stop0. It means that the layer counting is in good agreement with the value of the background accumulation rate. Second, at Stop0, the chronology also uses two tie-points from beta counting and gamma spectroscopy which are hence in agreement with annual layer counting and background accumulation rate. For Stop5, no beta counting and gamma spectroscopy has been used in the chronology construction but, as for Stop0, the annual layer counting is associated with small uncertainty and there are numerous tie-points ($nssSO_4$ and water isotopes) between the 3 cores. A third support for the chronology of the stack records come from the $nssSO_4$ increase which could be related to the Pinatubo eruption. Figure 2 shows the depth range corresponding to the period 1992-1993 on each core. At both Stop0 and Stop5, it corresponds to a depth range with a level of $nssSO_4$ which is relatively high.

At D47, there is a very poor agreement between the a priori and output accumulation rate when a 200% uncertainty is used for the background scenario. This reflects a disagreement between layer counting and background accumulation rate. We have however very low confidence in layer counting at D47 (see above). Note that we find correct dates for the beta counting and gamma spectroscopy peaks not used in the chronology construction when we use the chronology built using an uncertainty of 30 % on the background accumulation rate. This is however not enough to produce a robust chronology for this site.

Finally, Paleochrono calculates an uncertainty (1 σ) for the chronology of the three stacked records (Figure 5). This uncertainty is relatively small. 1 σ is less than 1 year for Stop0 and less than 1.5 year at Stop5 which reflects the good confidence in the layer counting and agrees with variability of the chronology when running sensitivity tests with Paleochrono (choice of tie-points, uncertainty on the background accumulation rate). At D47, 1 σ is less than 6 years which seems underestimated when comparing the chronology obtained with an





uncertainty of 30 or 200% on the background accumulation rate (Figure S3). This underestimation is linked to effects not yet implemented in Paleochrono such as the fact that the biases in background accumulation rate or layer counting for the neighboring cores are not independent.




**Figure 4: Resulting stacks for each site when the series are unmatched (in red) and when the matching is performed with Paleochrono with dated intervals but using no tie-points (in black), with nssSO₄ tie-points (in green), water isotope tie-points (in yellow) or both nssSO₄ and water isotopes tie-points (in purple). Shaded envelopes correspond to the minimum and maximum values of the 3 records used to build each stack. Note that for D47, the age scale displayed here has been obtained with a Paleochrono run with an uncertainty of 200% for the prior accumulation rate (the output for an associated background uncertainty of 30% is displayed on figure S6).**


### 3.5 Virtual firn cores



**Figure 5: Stacked data and associated dating uncertainty (1 σ) calculated by Paleochrono in black. The shaded envelopes of the stacked curves correspond to the minimum and maximum values of the individual records at each site. Blue and red curves are the δ¹⁸O records of virtual firn cores inferred from ECHAM6-wiso and LMDZ6iso outputs respectively.**



**Orange curve for Stop5 site corresponds to the δ¹⁸O composite curve after setting January 1ˢᵗ of each year as the date for the nssSO₄ peak. The depth resolution of the VFC is 5 mm.**

Figure 5 presents the $\delta^{18}O$ signal at the three sites as obtained from the virtual firn core calculation using either the ECHAM6-wiso or the LMDZ6iso model. In both cases, the virtual water isotopic records display clear seasonal cycles which persist despite diffusion in the firn.

In general, there is an excellent agreement in the variability of the water isotopic records of the VFC obtained using either the ECHAM6-wiso or LMDZ6iso output even if the mean $\delta^{18}O$ value is 5 ‰ lower using

LMDZ6iso than ECHAM6-wiso at Stop0 (Table S4). Finally, we observe that the seasonal cycles are less visible in the D47 VFC when compared to the VFC for Stop0 and Stop5. This is due to the significantly lower accumulation rate at D47.

**4 Discussion**

**4.1 Comparison of Virtual Firn Core and stack series**

We observe (Table S4) similar mean $\delta^{18}O$ values at D47 and Stop5 for both models (D47: -27.2 ‰ and -27.4 ‰; Stop5: -32.5 ‰ and -33.7 ‰ for ECHAM6-wiso and LMDZ6iso, respectively) compared to the stack of measured $\delta^{18}O$ records (D47: -26.9 ‰; Stop5: -34.1 ‰). The agreement is less good at Stop0 where we find significant differences between the mean $\delta^{18}O$ of the stack data (-36.7 ‰) and the modeled values (-34.2 ‰ for

ECHAM6-wiso and -39.5 ‰ for LMDZ6iso). The model-data discrepancies can at least partly be related to the difference in temperature between model outputs and measurements obtained from the automated weather data or borehole temperature (Table S4).

The amplitude of the seasonal cycles is in general lower in the stack than in the VFC, which is expected since stacking series tends to decrease the variability in case of slight misalignments. At Stop5, the mean seasonal

$\delta^{18}O$ amplitude is larger in the modeled series (5.1 ‰ and 5.8 ‰ for respectively ECHAM6-wiso and LMDZ6iso) than in the stacked series (3.2 ‰). The same is observed for Stop0 with values of respectively 5.4 ‰ and 9.9 ‰ for the mean seasonal cycle in the VFC based on the ECHAM6-wiso and LMDZ6iso models, and 3.6 ‰ for the stacked series. At D47, the amplitude of the variability is comparable between the modeled series (2.0 ‰ and 2.2 ‰ for ECHAM6-wiso and LMDZ6iso respectively) and the stacked series (2.7 ‰). This result is surprising as

D47 features strong katabatic winds that induce erosion and upper mixing not included in the VFC but which are expected to decrease the amplitude of the isotopic variability.

Despite the general good visual agreement between model and stacked ice core variability on the whole series, some discrepancies are visible over some sections. The upper part of the records appears much smoother in the stacked data than in the modeled VFC isotopic records. This cannot be attributed to the stacking itself removing

some of the real variability as individual cores also feature a reduced variability compared to the VFC (Figure S3, S4). The measurement technique is also not responsible for such smoothing effect because similar effects are seen when the measurements are performed through continuous flow analysis or on discrete samples. In the VFC isotopic records, the simulated diffusion follows the approach of Johnsen et al. (2000) and remains weak for the upper part of the core. However, surface firn is fragile and porous making it more susceptible to further diffusion



during transport and storage (Dallmayr et al., 2025a) not considered in the VFC construction. Finally, the upper part of the firn is also very sensitive to wind pumping, snow mixing and redistribution, especially at windy sites. It is difficult to discriminate between these three processes from the time series of isotopic composition alone, so we have decided not to account for them in the construction of the VFC signals. Still, they can explain the rapid loss of variability of the isotopic records at the top of the shallow cores (Casado et al, 2020).

At D47, we observe only 31 maximums in the stacked $\delta^{18}$O data while the record is expected to last 39 to 62 years from our chronology construction (Table 4), whereas on Stop5 and Stop0, we see one clear $\delta^{18}$O maximum every year except for a few years where we better see a shoulder in the $\delta^{18}$O record. The difference in the number of peaks between Stop0 – Stop5 and D47 is the combination of the significantly lower amount of accumulation at D47 compared to Stop0 and Stop5, as well as large amount of erosion or mixing of surface snow by wind (Picard et al., 2019; Zuhr et al., 2023) at D47 which is one of the places with the strongest katabatic winds in Antarctica (Davrinche et al., 2024; Kodama et al., 1985). This effect can be seen through the difference of the local roughness between the 3 sites (Table 4): the root mean square of surface deviation is 2.7 m at D47 compared to values of 0.06 m and 0.1 m respectively for Stop0 and Stop5. Such topographic effects were already shown to create $\delta^{18}$O variations which cannot be related to climatic variability (Dallmayr et al., 2025b). We thus conclude that D47 is not a good site to provide annually resolved climate and/or atmospheric water cycle reconstructions from water isotopic records in firn or ice core. In the following, our analysis focuses on the records obtained at Stop0 and Stop5 only.

**Table 4: Comparison of the timespan of the whole stacks from Paleochrono chronology (Figure S3), the observed number of $\delta^{18}$O maximums, the windspeed values from ERA5 reanalysis over the period 1979-2016 (mean, standard deviation and maximum value) and the root mean square of surface deviation on a circle of 300 m around the drilling as indication of the surface roughness obtained from the Reference Elevation Model of Antarctica (Howat et al., 2019). The two values given for the timespan of the stack for D47 correspond to the Paleochrono runs with respectively an uncertainty of 200% (lowest value) and 30% (highest value). The automatic weather station at D47 recorded wind speed over the period 2001 – 2017.**

| Site | Time span of the stack (yr) | Observed number of $\delta^{18}$O maximums | Mean windspeed (m/s) (ERA5) (max value) | Automatic weather station (m/s) | Root mean square of surface deviation (m) |
|---|---|---|---|---|---|
| **Stop0** | 47 | 45 | 7.6 ± 1.0 (13.8) | - | 0.06 |
| **Stop5** | 44 | 44 | 8.6 ± 1.1 (16.1) | - | 0.10 |
| **D47** | 39-62 | 31 | 12.1 ± 1.5 (21.8) | 12.3 ± 3.9 (32.5) | 2.78 |

At Stop5 and especially at Stop0, there is a strong resemblance between the stack and the VFC records. Still, we observe small temporal shifts (< 1 year), especially since our stack maximums do not always coincide with summer. This mismatch is within the uncertainty of the chronology and is inherently linked to the method used here for the chronology construction which does not impose any tie-points between peaks in $\delta^{18}$O or nssSO$_4$ and summer periods. Numerous studies however already used the nssSO$_4$ peaks as markers for austral summer when dating Antarctic ice cores (Emanuelsson et al., 2022; Steig et al., 2005). Indeed, the main source of nssSO$_4$





is dimethyl sulfide (DMS) produced by phytoplankton blooms starting in December with the break-up of sea ice
(Abram et al., 2013). Maxima of ☐$^{18}$O are also routinely used as markers for summer periods for firn and ice cores

when annual layers are clearly recorded (e.g. Emanuelsson et al., 2022; Vega et al., 2016). Here, we follow these
approaches and modify the chronologies obtained from Paleochrono by forcing the nssSO$_4$ peaks and/or the ☐$^{18}$O
maxima to coincide with summer periods. At Stop5, because the nssSO$_4$ profiles are complete, the alignment can
be performed through matching the nssSO$_4$ peaks from the stack record with the summer periods (1$^{st}$ of January)
starting from the most recent period (Figure S5). When plotting the stack of water isotopes on this new chronology,

we find a good match with the VFC ☐$^{18}$O record with both maxima in the stack ☐$^{18}$O and maxima in the VFC
☐$^{18}$O records occurring during the austral summers. At Stop0, because of the lack of nssSO$_4$ data, we matched the
☐$^{18}$O maxima with maxima in the VFC ☐$^{18}$O records (list of tie points in Table S5) resulting in a modification of
the stack chronology by less than 1 year which does not significantly modify the uncertainty obtained from layer
counting. Note that when we do the matching through ☐$^{18}$O variations at Stop5, we obtain similar results than for

the matching through nssSO$_4$ maximums (Figure 5).

Table S6 compiles the correlation coefficients between individual cores or stacked data at each site and
VFC $\delta^{18}$O series after modification of the chronology to have nssSO$_4$ and ☐$^{18}$O peaks during summer. We find
correlation coefficients $r$ between the stack and the VFC of 0.58 and 0.54 (ECHAM6-wiso) and 0.55 and 0.60
(LMDZ6-iso), at Stop0 and Stop5 respectively which further motivate using such stacks to try to recover climatic

information in the following. Because both LMDZ6iso and ECHAM6-wiso VFC models display similar variability
in the VFC, we only show analyses performed with ECHAM6-wiso VFC in the following section. Same
conclusions can be drawn from LMDZ6iso VFC.

### 4.2 Identification of climate signals in the stack

In a previous study performed on a single firn core in coastal Adélie Land, Goursaud et al. (2019) found
no clear correlation between $\delta^{18}$O and temperature at the seasonal scale. Later, Leroy Dos Santos et al. (2023)
provided 2 years of atmospheric monitoring of water vapor $\delta^{18}$O at Dumont d'Urville station and showed that if
$\delta^{18}$O of water vapor is not strongly correlated to temperature, there is an excellent agreement between the $\delta^{18}$O of
water vapor modeled by the ECHAM6-wiso model and the one measured on-site. This result, together with the
good performance of the ECHAM6-wiso model when simulating climatic parameters, shows that the ECHAM6-
wiso model properly reproduces the climatic signature of the water vapor isotopic signal. The agreement was less
good for the precipitation $\delta^{18}$O. When using the VFC $\delta^{18}$O signal calculated from ECHAM6-wiso outputs, the
agreement was poor with the $\delta^{18}$O signal measured in single firn core (Leroy Dos Santos et al., 2023). One of the
reasons invoked for this mismatch was stratigraphic noise. In our study, we can test this hypothesis since we
removed at least part of the stratigraphic noise by stacking three records at each site. In the following, we thus test
how our stacks can be used to get representative insights of the regional climate over the past decades.

Since both the VFC signals and stack records display clear seasonal cycles and interannual variability,
we concentrate on the possible identification of extreme warm summer in our records. First, we address the
possible link between the extreme values in $\delta^{18}$O and in temperature. To infer how the stacks can be used to identify
years with particular/extreme climatic patterns, we compare how the 12 warmest summers of the period 1979 –
2016 can be identified in the stacks. This comparison is done for Stop0 where we find the best visual agreement
(and the highest correlation) between the stack and VFC ☐$^{18}$O records. Table 6 (right part) indicates the 12 years



with the warmest summer periods according to the ERA5-derived temperature. From these, only 6 are also within the 12 highest summer peaks in the stack $\delta^{18}O$ curve. Even considering a possible uncertainty of our chronology by 1 or even 2 years, we cannot match the 12 warmest summer periods with the 12 highest summer peaks in the stack $\delta^{18}O$ curve. This confirms that the stack $\delta^{18}O$ curve cannot be used to robustly identify the extreme years in terms of temperature variations. This is not unexpected since, as mentioned above, $\delta^{18}O$ is not a direct indicator of temperature and cannot directly be used to identify maximum or minimum temperature levels; it is also influenced by the seasonality of the precipitation, the precipitation intermittency, the climate at the evaporative source and the trajectory of the vapor precipitated.

To address the issue of the intermittency of the precipitation, we then compare the 12 highest peaks in precipitation-weighted monthly temperatures with the peaks in the stack $\delta^{18}O$ and only find 7 peaks in common (Table 6, left part). This means that precipitation intermittency is not the only reason why $\delta^{18}O$ archived in snow and ice in coastal Adélie Land is not a perfect indicator of past temperature. A second limitation is linked to the influence of atmospheric processes within the water cycle and atmospheric transport on the isotopic composition of precipitation. To account for this effect, we compare the VFC $\delta^{18}O$ record with the stack $\delta^{18}O$ record. In this case, 8 of the 12 most prominent $\delta^{18}O$ peaks in the VFC isotopic curve are among the 12 most prominent $\delta^{18}O$ summer peak in the stack record (Table S7). Again, it is not possible to match the 12 highest $\delta^{18}O$ maxima of the VFC and the stack within the dating uncertainty.

**Table 6: Summers associated with the 12 highest monthly precipitation-weighted temperatures at Stop0 from ERA5 (1ˢᵗ column), and with the 12 highest monthly mean temperatures (4ᵗʰ column). For each of these years, we indicate a cross if this year also corresponds to one of the 12 maximums in the stacked $\delta^{18}O$ series (2ⁿᵈ and 5ᵗʰ columns) and in the VFC $\delta^{18}O$ series (3ʳᵈ and 6ᵗʰ columns), and in orange shading the years were all the series record one of the 12 highest values. (Same analysis with LMDZ6is-derived VFC is presented in Table S8).**

| Maximums in summer precipitation weighted monthly temperature | Maximums in stack $\delta^{18}O$ | Maximums in ECHAM6-wiso VFC $\delta^{18}O$ series | Maximums in summer monthly temperature | Maximums in stack $\delta^{18}O$ | Maximums in ECHAM6-wiso VFC $\delta^{18}O$ series |
|---|---|---|---|---|---|
| 2013-2014 | X | X | 2013-2014 | X | X |
| 2011-2012 | X | X | | | |
| 2010-2011 | X | | | | |
| | | | 2009-2010 | | X |
| 2005-2006 | X | | 2005-2006 | X | |
| 2004-2005 | | X | 2002-2003 | | X |
| 2001-2002 | X | X | 2001-2002 | X | X |
| 1993-1994 | | | 2000-2001 | | |
| 1991-1992 | | | 1991-1992 | | |
| 1989-1990 | | | | | |
| 1985-1986 | X | X | 1986-1987 | X | X |
| 1984-1985 | X | X | 1984-1985 | X | X |
| 1983-1984 | | | 1983-1984 | | X |



| | | | 1982-1983 | | |
|---|---|---|---|---|---|
| | | | 1979-1980 | X | |
| **Proportion** | 7/12 | 6/12 | **Proportion** | 6/12 | 7/12 |

If stacking 3 cores at one site is expected to remove the stratigraphic noise and increase the resemblance between the isotopic records and the VFC $\delta^{18}O$, discrepancies remain which can be attributed, at least partly, to the variability from one core to the other (to be attributed to deposition or post-deposition effects). Except for four very prominent summer periods identified in Table 6 for stack $\delta^{18}O$, temperature, weighted temperature and VFC $\delta^{18}O$, the variability in the seasonal amplitude of the $\delta^{18}O$ record among individual firn core leads to a muted stack

signal with a relatively large uncertainty envelope. Stacking more than 3 cores at one site may help reducing this envelope and better resolve the interannual variability. Another reason for the discrepancy between stacked $\delta^{18}O$ and VFC $\delta^{18}O$ series could also be that the models do not faithfully reproduce the interannual variability in $\delta^{18}O$ of precipitation as observed in Leroy-Dos Santos et al., 2023. In particular, the AGCM may not fully reproduce the amplitude of the synoptic events which strongly contribute to the $\delta^{18}O$ signal in coastal Adélie Land (Leroy-

Dos Santos et al., 2023).

     Despite these limitations in faithfully inferring the interannual variability of the $\delta^{18}O$ from a stack of 3 cores at the same site, the overall good resemblance between the stack and the VFC isotopic curves support the use of such stacked cores to reconstruct first order trends in $\delta^{18}O$ over the last decades or centuries. At Stop5 and Stop0, the overall good agreement between the stack and the VFC (Figure 5), in spite of the unavoidable dating

uncertainty, is a promising result. At coastal sites, where pre- and post-depositional processes significantly amplify $\delta^{18}O$ variability, reconstructions of $\delta^{18}O$ series from stacking of several records are thus crucial. They enhance the representativeness of the site, may improve our understanding of the processes driving non-climatic $\delta^{18}O$ variability, and serve as valuable validation data for climate models. Such results open the door toward model-data comparison in coastal Antarctica to better understand how much climate signal is preserved and archived in

coastal ice and firn cores.

**5 Conclusion**

     We present an analysis of 9 firn cores drilled at three sites (3 cores at each site) across the coast-to-plateau transition in Adélie Land. These cores, all spanning at least the recent period from 1979 to 2016, were analyzed at high resolution for water isotopes and impurities using a continuous flow analysis set-up. Such high-resolution

measurements in a region of high snow accumulation rate permit to capture the seasonal variations both in the chemistry and water isotopic records. These variations are key to produce a chronology for the different firn cores. At each site, we use the similarities in the $nssSO_4$ and $\delta^{18}O$ variations to stack the three isotopic records in one curve hence increasing the signal-to-noise ratio by getting rid of some stratigraphic noise. A comparison of these isotopic records [with] VFC isotopic curves produced from outputs of the two AGCMs ECHAM6-wiso and

LMDZ6iso shows that stacking does not enable the recovery of a climatic signal at the very windy place of D47 but helps reconstruct first order trends in $\delta^{18}O$ variability at other sites of coastal Adélie Land with lower wind speed and weaker erosion processes.



In this study, we used this consistency between the VFC and the stacked $\delta^{18}O$ profiles to compare the relative amplitude of summer temperature and $\delta^{18}O$ in both the AGCMs outputs and our stack $\delta^{18}O$ profiles. This analysis confirms the complexity of linking temperature and $\delta^{18}O$ in this region, as suggested by previous studies but permits to identify some extreme summers. To improve such reconstruction, a next step is probably to produce stack records from a larger number of firn cores to reduce the uncertainty envelope associated with inter-core variability and obtain a more faithful reconstruction of $\delta^{18}O$ variability at a same site. It is also necessary to further investigate how climate information underlies the good correlations observed between the stacks and the VFC. Such an exercise could potentially help identify extreme years in series of $\delta^{18}O$ over the last decades to centuries but requires an important analytical load.

**Data availability**

All the data used in this publication can be found at Zenodo (https://doi.org/10.5281/zenodo.15672732). It includes $\delta18O$, $\delta D$ and chemical impurities data for the individual cores at all sites, stacked $\delta18O$ and $\delta D$ data along with their associated chronologies and the $\delta18O$ data of the Virtual Firn Cores built from both ECHAM6-wiso and LMDZ6iso.

**Author contributions**

Conceptualization: TT, EF, VF and AL. Methodology: TT, EF, CLDS, FPr, ELM, FPa, OJ, RJ, BM, OM, CA, ND, VF, LB, CLB, MC, MW, AC, LA, BJ, GP and MB. Investigation: TT, EF and AL. Writing original draft: TT, EF, AL Writing– review and editing: EF, AL, FP, ELM, FPa, OM, CA, ND, VF, MC, MW, AC and GP.

**Competing interests**

The contact author has declared that none of the authors has any competing interests.

**Acknowledgements**

The authors *acknowledge* the support from Agence Nationale de la Recherche, projects *ANR-20-CE01-0013 (ARCA), ANR-14-CE01-0001 (ASUMA), the LEFE-IMAGO program ADELISE as well as the IPSL project TADAM. The authors also acknowledge the AMRC (Antarctic Meteorological Research Center) based at the University of Wisconsin for supporting and maintaining the Automatic Weather Station (AWS) network, and the Institut Paul-Emile Victor for the IPEV project ADELISE (1205) as well as its logistical support in the field of the Antarctic part of the French glacier inventory* (stake data of the program *GLACIOCLIM—SAMBA*) and in particular for the maintenance of *D47 AWS.*



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
