# Peer review of "Multiproxy analyses of multiple firn cores from coastal Adélie Land covering the last 40 years"

_EGUsphere, 2025_

## Author Comment (AC1)

**Response to Reviewer's comment 1 (RC1)**

We thank reviewer 1 for its positive feedbacks on the manuscript and very helpful comments and suggestions to improve it. We address below the comments with a point-to-point answer.

RC1: 'Comment on egusphere-2025-2863', Anonymous Referee #1, 02 Nov 2025

The paper by Tcheng et al. is reporting an in-depth analysis of the isotopic composition of individual and stacked records from firn cores (a total of 9 cores) collected at three sites in a coastal area of Adelie Land in Antarctica. These sites are characterized by a relatively high snow accumulation rate allowing, through a continuous flow analysis method, to obtain high resolution d18O records along with nssSO4 profiles. These high-resolution records allow a very precise dating, at least for the two sites named Stop5 and Stop0. All the cores exhibit seasonal variations but at the D47 site the strong katabatic winds are strongly affecting the isotopic records. A stacking record, obtained for each of the three sites, permits to reduce the stratigraphic noise and is then compared to a Virtual Firn Core (VFC) calculated using two isotopic GCMs, ECHAM6 and LMDZ6, over a common period 1979-2016. The two less disturbed sites show consistent seasonal variability as the VFC records, allowing to observe common trends in d18O. This study, through a very detailed analysis if the high-res records, demonstrate the difficulty in obtaining a clear link between temperature and d18O, particularly in those sites that may be disturbed by redistribution processes by winds, as is the case for D47.

The paper is original and novel in the sense it uses a very detailed isotopic and chemical records for obtaining a climate reconstruction to be comparable to the climate data obtained from isotopic models. The data are well presented and discussed, and I have found the reading quite smooth although some parts could be reduced a little.

I recommend its publication after the authors have been considered to the following minor comments.

I have a general comment: why not attempting to calculate mean annual d18O values to be compared to VFC data and ERA5 precipitation weighted data? This could bring some information to eventual the d18O/T sensitivity at interannual scale. Has this been done or at least checked?

>> Many thanks for this comment. Indeed, we checked how it compares and unfortunately, the comparison is not very convincing. When looking at the temporal evolution of mean annual stacked $\delta^{18}O$ and weighted temperature (Figure 1), we do not observe a very strong correlation which is confirmed by the graph showing the evolution of stacked $\delta^{18}O$ vs weighted temperature (Figure 2).

[Figure]

***Figure 1**: Temporal evolution of the mean annual weighted temperature (red) and stacked $\delta^{18}O$ (blue) at stop0. Averages are calculated from January to December (tests were also performed when calculating the average from July to June leading to the same results and conclusions).*

[Figure]

***Figure 2**: Relationship between mean annual stacked $\delta^{18}O$ and weighted temperature over the period 1979 - 2016. Averages are calculated from January to December (tests were also performed when calculating the average from July to June leading to the same results and conclusions).*

The title: I would suggest changing "covering the last 40 years" since it is misleading, they are not the last 40 years…

>> The title has been changed as "**Multiproxy analyses of multiple shallow firn cores from coastal Adélie Land**"

Page 2, line 1: " … closely linked to …. ice mass loss …"

>> Corrected

Page 2, line29: may you specify "long-term" how many years it is?

>> We already specified the period in the submitted version: "(continuously since January 2019)"

Page 7, line 147: may you add which is the period covered?~l

>> Many thanks, we now precise "(~last 40 years)". We can not be more precise since this sentence is written before building the chronology.

Page 10, line 232: all the nssSO4 profiles feature common patterns. However, If I am looking at figure 2 the sulphate records at Stop5 are not so similar….

>> Our sentence was probably not very clear. We referred to the similarities between nssSO4, Na and Cl records on each core. The sentence has been rewritten as "For each core, the variability recorded in the $nssSO_4$ profiles (Figure 2 and S1) share strong similarities with those in $Na^+$ or $Cl^-$ profiles˝

Page 10, lines 248-249: the Pinatubo signal: I agree with the authors that it is quite ambiguous, but if I am looking at the figure S5 the signal is well evident at Stop0 (stacked profile).

>> Many thanks for your comment. As mentioned, we can see a signal corresponding to this eruption ("A slight increase in the average value of $nssSO_4$ can still be observed between 10 and 12 m at Stop0 and Stop5 and between 7 to 12 m at D47 and may include the signature of the Pinatubo eruption "). Still, because it was quite ambiguous, it was not used for the chronology construction. It is however a good support for the dating that we have this small signal at the right age.

Page 13, Table 3 and also in the text to be specified if the data regarding the snow accumulation from stake is referring to a mean value obtained from a stake farm or is one single stake value and which density values have been used and to which year is referred.

>> At Stop0, 5 stakes are used and the accumulation is calculated over the period 5/12/2016 to 30/1/2020. We added the number of stakes used at Stop0 in the caption and in the table. The density used at stop0 is the one measured on the firn core.
At D47, there are 2 stakes (as written already in the first manuscript) and the value is given as an average of annual accumulation estimated each year over the period 2004 - 2020. The density is measured every year through a 250 cm profile.
All missing details will be given in the revised manuscript.

Page 18, figure 5: If I am looking at the records from Stop5 I found very different trends and patterns between staked records and VFC ones, around 1990 +/-3 years. May you comment on this? Perhaps are you referring at this point at page 19 lines 392-393? Add in the text.

>> We propose to add the following sentence "As an example on Stop5, we see that while the stacked data displays a clear maximum around year 1989, the modeled signal is much more muted. "

Page 20, lines 411-413: may you explain better for the reader how the local roughness is calculated. There is an explanation in the table caption, but I would suggest you move it to the main text.

>> Done. We added the following sentence "the surface roughness is documented by calculating the root mean square of surface deviation on a circle of 300 m around the drilling as obtained from the Reference Elevation Model of Antarctica (Howat et al., 2019). "

Page 20, lines 427-429: see my previous comment above (figure5).

>> We have removed the definition from the caption.

Please, change in all the figures (text and supplementary) the X axis title and labels from mm to m.

>> Corrected

Please check all the delta symbols in the text.

>> Indeed, some "delta" were not well formatted. This will be addressed when sending the final manuscript through a thorough final reading.

Please check all the table format.

>> Corrected

In the Supplementary: the caption of figure S5 is referring to which core site? Stop0?

>>It was Stop5. We now specify the site in the caption

Is the code for calculating the VFC records free available?

>> The code can be found here: https://gitlab.in2p3.fr/ipsl/lsce/glaccios/glacio-psm
It will be mentioned in the revised version of the manuscript.